# Meta-analytic evidence that sexual selection improves population fitness

Justin G. Cally [1], Devi Stuart-Fox[1] & Luke Holman[1]

Sexual selection has manifold ecological and evolutionary consequences, making its net effect on population fitness difficult to predict. A powerful empirical test is to experimentally manipulate sexual selection and then determine how population fitness evolves. Here, we synthesise 459 effect sizes from 65 experimental evolution studies using meta-analysis. We find that sexual selection on males tends to elevate the mean and reduce the variance for many fitness traits, especially in females and in populations evolving under stressful conditions. Sexual selection had weaker effects on direct measures of population fitness such as extinction rate and proportion of viable offspring, relative to traits that are less closely linked to population fitness. Overall, we conclude that the beneficial population-level consequences of sexual selection typically outweigh the harmful ones and that the effects of sexual selection can differ between sexes and environments. We discuss the implications of these results for conservation and evolutionary biology.

---

[1] School of BioSciences, The University of Melbourne, Parkville, VIC 3052, Australia. Correspondence and requests for materials should be addressed to J.G.C. (email: justin.g.cally@gmail.com)

Sexual selection, defined as selection resulting from competition for mates or their gametes[1], is a ubiquitous evolutionary force that has profoundly shaped the natural world. As far back as Darwin[2], researchers have theorised that sexual selection can change the average absolute fitness of individuals in a population, henceforth termed 'population fitness'[3]. However, opinion is divided over whether the net effect on population fitness is positive or negative[4–7]. Prima facie, one might predict that sexual selection would have no effect on population fitness, since it does not matter which individuals of the faster-reproducing sex (typically males) succeed in mating, so long as some do[8]. However, when genotypes with high mating or fertilisation success also have superior breeding values for traits that affect population fitness (e.g. survival, parental care, female fecundity or success in interspecific competition), sexual selection is predicted to elevate population fitness by causing a correlated response in these other traits[7]. In essence, the demographically limiting sex (typically females) benefits from a gene pool that has been purged of harmful alleles through sexual selection on the non-limiting sex (typically males). Theoretically, the benefit to population fitness could be large[9–11].

Conversely, sexual selection can decrease population fitness if male sexually selected traits are negatively genetically correlated with female fitness, producing intralocus sexual conflict[12–16]. Additionally, sexual selection frequently favours phenotypes that reduce population fitness but benefit the individuals expressing them, such as harassment or infanticide by mate-seeking males[17], as well as investment in costly sexual signals and weaponry at the expense of parental care: *inter*locus sexual conflict[18,19]. Given these conflicting theoretical expectations and empirical results, it remains unclear whether sexual selection tends to have a net benefit or cost to population fitness[4–7].

Researchers have investigated the population-level consequences of sexual selection using a range of approaches including macro-evolutionary studies[20–22], analysis of the fossil record[23], quantitative genetics[12,24–27] and especially experimental evolution. In particular, many experimental evolution studies have manipulated the intensity of sexual selection in captive populations, allowed evolution to proceed and then measured population fitness components such as lifespan, reproductive success, population extinction rate and mutation load. This approach facilitates direct measurement of the net effect of sexual selection on population fitness, at least in the specific populations and ecological conditions under study.

A number of factors might influence the strength and sign of the correlation between sexual selection and population fitness. First, the genetic correlation between female fecundity and male mating/fertilisation success varies in sign and magnitude between species[28] and even between conspecific populations[29], implying that sexual selection on males increases mean female fitness in some species and populations but not others. These inconsistencies could derive from differences in allele frequencies, or environmental differences that alter how genotype relates to phenotype and fitness. Second, it has been hypothesised that populations should display a more positive genetic correlation between male and female fitness—and thus potentially between mating/fertilisation success and population fitness—in novel or fluctuating environments, relative to stable environments[12,27,30,31]. This is because stable environments create consistent selection, preferentially eroding genetic variation at sexually concordant loci (i.e. loci where the fittest genotype is the same in both sexes) and leaving behind variation at sexually antagonistic loci. We know of no systematic reviews of this latter theory, though it is has motivated several recent empirical tests[12,24,26], and is relevant to conservation genetics.

Here, we synthesise the empirical literature on sexual selection and population fitness using formal meta-analysis. We focus exclusively on experimental evolution studies that manipulated the presence or strength of sexual selection on males, and then measured some fitness component, since experiments provide a particularly strong test of the hypothesis that sexual selection affects the average fitness of populations. We found that sexual selection tends to improve population fitness, especially when fitness components were measured in females experiencing stressful rather than benign conditions. Additionally, we show that sexual selection tends to narrow phenotypic variance of fitness-related traits for females and mixed-sex samples in stressful conditions. These results suggest that sexual selection may be especially important for populations adapting to changing environments.

## Results

**The effect size dataset**. We retrieved 459 effect sizes from 65 studies. Ninety two effect sizes were collected from populations evolving under stressful conditions, while 337 were measured on populations evolving in benign conditions. One hundred and eighty nine of the effect sizes came from measurements made on males, 219 on females and the remaining 51 from measurements of a mixed-sex sample of individuals. Most effect sizes in our dataset came from studies that manipulated sexual selection by completely removing it in one treatment via enforced random monogamy ($n = 241$); other effect sizes ($n = 218$) were derived from alternative manipulations, such as changing the adult sex ratio. In total, we obtained effect sizes for 22 different fitness traits, with female reproductive success ($n = 102$) and offspring viability ($n = 56$) being the most commonly measured traits. We classified 171 effect sizes as direct measures of population fitness, 141 as indirect and the remaining 144 effect sizes as ambiguous (see Methods, Fig. 1 and Supplementary Table 1). Specifically, we scored traits that are likely to correlate with population growth and persistence as direct (e.g. female reproductive success, offspring viability and extinction rate), those which are not necessarily correlated with population fitness but which do measure individual fitness as indirect (e.g. lifespan, male mating success and ejaculate quality/production) and those for which the relationship to population fitness is unclear as ambiguous (e.g. body size, mating duration, male reproductive success and early fecundity). Supplementary Tables 3 and 4 give a detailed description of our dataset.

**Sexual selection is associated with higher mean values for most fitness components**. The grand mean across all types of effect sizes (direct, indirect and ambiguous) was positive (restricted maximum likelihood (REML) $\beta = 0.24$, 95% confidence intervals (CIs): 0.055–0.43, $p = 0.011$; Bayesian $\beta = 0.25$, 95% CIs: −0.0074 to 0.51, $BF_{>0} = 35$), indicating that sexual selection on males typically had a net positive effect on the majority of populations and fitness components so far studied. Moreover, the effect sizes associated with the manipulation of sexual selection varied between different fitness traits. Sexual selection had a beneficial effect on most fitness traits, but varied across the three relationships to fitness (Fig. 1; Supplementary Tables 5 and 6). Sexual selection elevated fitness for traits that shared an ambiguous relationship to fitness (REML $\beta = 0.21$, 95% CIs: 0.058–0.093; Bayesian $\beta = 0.20$, 95% CIs: −0.0016 to 0.39, $BF_{>0} = 38$, $n = 144$) and an indirect relationship to fitness (REML $\beta = 0.24$, 95% CIs: 0.13–0.36; Bayesian $\beta = 0.24$, 95% CIs: 0.033–0.43, $BF_{>0} = 59$, $n = 141$). Additionally, sexual selection elevated fitness components directly related to fitness, albeit at a lower magnitude (REML $\beta = 0.13$, 95% CIs: 0.019 to 0.24; Bayesian $\beta = 0.13$, 95%

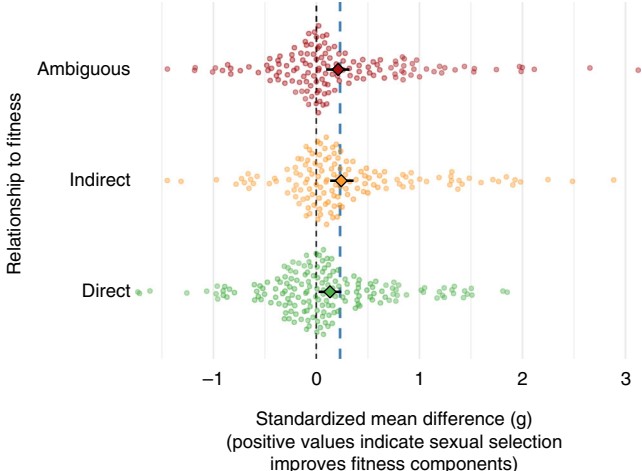

**Fig. 1** The effect of sexual selection on direct, indirect and ambiguous fitness components. The effect sizes used in this meta-analysis ($n = 459$) were grouped into either direct, indirect or ambiguous measures of fitness. Overall, effect sizes were more often positive than negative. Predicted average values are presented as a diamond with 95% confidence intervals (CIs) for each fitness-relationship category. The estimates presented here are from restricted maximum likelihood (REML) models with the grand mean across all effect sizes ($\beta = 0.25$) shown as the blue dotted line. Predictions from both Bayesian and REML models can be found in Supplementary Table 6. Data used for generating this figure are available in the Source Data file

CIs: −0.079 to 0.31, $BF_{>0} = 11$, $n = 174$). A large forest plot with predicted effect sizes for each fitness component is presented in Supplementary Fig. 1 and further detailed with model predictions and individual meta-analyses in Supplementary Tables 7 and 8. Sexual selection significantly reduced two fitness components, namely immunity (REML $\beta = -0.42$, 95% CIs: −0.64 to −0.20; Bayesian $\beta = -0.43$, 95% CIs: −0.70 to −0.15, $BF_{>0} = 0.0026$; $n = 35$) and body condition (REML $\beta = -1.2$, 95% CIs: −1.8 to −0.63; Bayesian $\beta = -1.2$, 95% CIs: −1.9 to −0.63, $BF_{>0} < 0.0001$, $n = 1$).

**The roles of environmental stress and sex**. We found that the sex of the individuals measured (male, female or a mixture) and the conditions under which the population evolved (stressful or benign) interacted to affect the relationship between sexual selection and fitness (Tables 1 and 2 and Supplementary Tables 11–13). Sexual selection on males significantly improved female fitness, and the beneficial effect of sexual selection was significantly stronger for females from populations evolving under stressful conditions (e.g. a food source to which they were not well adapted) than under benign conditions (Fig. 2a, Table 2, Supplementary Table 14). Sexual selection had a positive but non-significant effect on male fitness, and in contrast to females, fitness benefits were significantly weaker in stressful than benign environments (Fig. 2a, Table 2). Consistent with the different consequences of sexual selection for female and male fitness, the mean effect size in mixed-sex samples was non-significantly positive, and there was no significant difference between benign versus stressful conditions (Fig. 2a, Table 2). Overall, our results indicate that the positive effect of sexual selection on fitness is greater for females than males, and the difference between the sexes is magnified in stressful environments. When only fitness components directly related to fitness were used in the mixed-effects model, the benefits of sexual selection were still magnified

in females evolving under stressful conditions (Supplementary Fig. 3 and Supplementary Table 15). Similarly, when we used an alternative measure of effect size (log response ratio or lnRR), the results aligned with those of Hedges' $g$ in that sexual selection elevates population fitness, with its effect magnified for females evolving in stressful environments (Supplementary Fig. 4 and Supplementary Table 16).

Other moderator variables that we examined had minimal impacts on effect size (Supplementary Table 9). Specifically, effect size did not depend on whether or not the study was conducted blind (Supplementary Fig. 7), nor on the number of generations for which the experimental evolution study was run (Supplementary Figs. 8 and 9).

The effect size estimates we recovered were highly heterogeneous ($I^2 = 95.2\%$, 95% CIs: 94.4–95.9), reflecting the large differences in experimental procedures, study species and fitness components included in our meta-analysis[32,33]. Heterogeneity stemmed mostly from between-study differences ($I^2_{study} = 36\%$, 95% CIs: 26.5–45.4) rather than differences between fitness components and taxon ($I^2_{fitness\ components} = 0.4\%$, 95% CIs: 0.2–0.9; $I^2_{taxon} = 1.4\%$, 95% CIs: 0.2–3.5). Variation among taxa is explored further in the Supplementary Information (Supplementary Fig. 2 and Supplementary Table 10).

**Sexual selection reduces phenotypic variance, for female traits in stressful environments**. By applying meta-analysis to log coefficient of variation ratios[34], we found evidence that sexual selection reduces phenotypic variation under certain conditions (Fig. 2b). Specifically, phenotypic variance was significantly reduced by sexual selection for fitness components measured in females under stressful conditions (log coefficient of variation (lnCVR) = −0.78, 95% CIs: −1.23 to −0.34, $n = 27$). By contrast, we found no significant effect of sexual selection on phenotypic variance in males, or for either sex under benign conditions (Fig. 2b; Supplementary Tables 17–20). However, similar to the results in females, there was a non-significant trend for a reduction in phenotypic variance in mixed-sex samples measured under stressful conditions (lnCVR = −0.76, 95% CIs: −1.22 to −0.31; Fig. 2b). A meta-analysis using the log variability ratio (lnVR), which does not account for the mean-variance relationship present in the dataset (Supplementary Fig. 5), suggested sexual selection reduces variance for mixed-sex samples in stressful conditions, but not females. Results of this meta-analysis can be found in the Supplementary Information (Supplementary Fig. 6 and Supplementary Table 21).

As in the meta-analysis of trait means, there was high heterogeneity in the estimates of lnCVR ($I^2 = 98.9\%$, 95% CIs: 98.7–99.1). Heterogeneity in the dataset was due to variability between studies fitness components ($I^2_{fitness\ components} = 12.7\%$, 95% CIs: 4.5–23.1) and taxon ($I^2_{taxon} = 7.7\%$, 95% CIs: 1.3–18.4), as well as large amounts of residual heterogeneity (78.6%, 95% CIs: 65.9–89). Using the REML approach heterogeneity associated with the Study ID random effect was estimated at zero ($I^2_{study} = 0\%$); however, the Bayesian approach suggests that heterogeneity between studies may be zero or small ($sd_{study} = 0.07$, 95% CIs: 0–0.19).

**Publication bias**. The funnel plot of effect sizes was asymmetrical, suggesting that some publication bias might be present (Fig. 3a; Egger's test: $z = 5.9$, $p < 0.0001$). Specifically, there was a moderate excess of low-powered studies in which sexual selection had a more positive effect on the fitness component than average, implying that low-powered studies are more likely to be published if they report statistically significant fitness benefits of

**Table 1 Multilevel meta-analysis model results**

| Parameters | Estimate[a] | SE | LCI | UCI | z | P value |
|---|---|---|---|---|---|---|
| REML model | | | | | | |
| Intercept | 0.19 | 0.12 | −0.05 | 0.43 | 1.53 | 0.13 |
| Both sexes | 0 | 0.07 | −0.14 | 0.15 | 0.04 | 0.97 |
| Female sex | **0.11** | 0.03 | 0.05 | 0.17 | 3.73 | 0.00 |
| Stressed environment | **−0.16** | 0.04 | −0.24 | −0.07 | −3.63 | 0.00 |
| Both sexes × stressed environment | **0.18** | 0.09 | 0.01 | 0.36 | 2.07 | 0.04 |
| Female sex × stressed environment | **0.26** | 0.05 | 0.16 | 0.37 | 5.11 | 0.00 |
| Bayesian model | | | | | | |
| Intercept | 0.188 | 0.18 | −0.17 | 0.52 | | |
| Both sexes | 0.003 | 0.07 | −0.14 | 0.14 | | |
| Female sex | **0.113** | 0.03 | 0.05 | 0.17 | | |
| Stressed environment | **−0.156** | 0.04 | −0.24 | −0.07 | | |
| Both sexes × stressed environment | **0.182** | 0.09 | 0.01 | 0.35 | | |
| Female sex × stressed environment | **0.264** | 0.05 | 0.16 | 0.37 | | |

Moderator variables whose 95% confidence intervals do not cross zero are shown in bold
REML restricted maximum likelihood, LCI 95% lower confidence interval, UCI 95% upper confidence interval
[a]Hedges' g is the response variable for this model

**Table 2 Hypothesis tests showing how sex and environmental stress interact to modulate effect size**

| Condition | Test | Statistical approach | Estimate[a] | SE | LCI | UCI |
|---|---|---|---|---|---|---|
| In benign environments | Female > male | REML | **0.113** | 0.03 | 0.05 | 0.17 |
| | | Bayesian | **0.113** | 0.03 | 0.05 | 0.17 |
| In stressful environments | | REML | **0.377** | 0.05 | 0.29 | 0.47 |
| | | Bayesian | **0.377** | 0.05 | 0.29 | 0.47 |
| For Females | Stressful > benign | REML | **0.108** | 0.04 | 0.04 | 0.18 |
| | | Bayesian | **0.109** | 0.04 | 0.04 | 0.18 |
| For Males | Benign > stressful | REML | **0.156** | 0.04 | 0.07 | 0.24 |
| | | Bayesian | **0.156** | 0.04 | 0.07 | 0.24 |
| For Both | Stressful > Benign | REML | 0.028 | 0.08 | −0.13 | 0.18 |
| | | Bayesian | 0.026 | 0.08 | −0.13 | 0.18 |

Moderator variables whose 95% confidence intervals do not cross zero are shown in bold
REML restricted maximum likelihood, LCI 95% lower confidence interval, UCI 95% upper confidence interval
[a]Hedges' g is the response variable for this model

sexual selection (though funnel plots are not a decisive evidence of publication bias[35]). Linear regressions show no significant relationship between effect size and journal impact factor (Fig. 3b; $t_{437} = 1.2$, $p = 0.23$) or year of publication (Fig. 3c; $t_{437} = −1.2$, $p = 0.24$); thus, we found no evidence that effect size dictates the likelihood of publication in high-profile journals, or that effect sizes have diminished as the field has matured[36,37].

## Discussion

Our meta-analysis revealed that populations evolving under sexual selection often have higher values for multiple fitness traits, relative to populations where sexual selection on males was experimentally removed or weakened. Sexual selection had beneficial effects on the majority of commonly measured individual fitness traits, which have indirect or unclear relationships to population fitness. Effects of sexual selection on direct measures of population fitness such as extinction rate, reproductive success (defined as the number of offspring produced) and the proportion of viable offspring were smaller and variable (similar to the results of another meta-analysis[38]), but nonetheless tended to be positive. Fitness traits related to immunocompetence were an exception: sexual selection typically resulted in weaker immunity. This result is interesting in light of the hypothesised trade-off between sexually-selected phenotypes and immunity, for example, due to immunosuppressive effects of sex hormones[39,40].

Furthermore, the overall benefit of sexual selection was greater for females than males, and this sex difference was magnified in stressful environments. Consistent with stronger selection on female fitness under stress, female and mixed-sex samples showed reduced phenotypic variance when sexual selection was applied under stressful as opposed to benign conditions. These results suggest that sexual selection may contribute to population persistence under stressful conditions, such as fluctuating environmental change[30] or spatial variability[41], particularly since female reproductive output is often a limiting factor in population growth[42].

The results of the meta-analysis support predictions that sexual selection on males can improve population fitness and accelerate adaptation[9–11,30,43,44]. One possible mechanism is that male mating success is positively genetically correlated with traits that contribute to population fitness, allowing females to benefit from a genome that has been purged of deleterious alleles through competition between males[7,9,10]. A second (non-exclusive) mechanism is that experimental manipulation of sexual selection on males might directly alter the selective pressures acting on females, causing female traits to evolve. For example, removing sexual selection via enforced monogamy probably alters selection on females, because it alters the frequency of interactions with males (as well as the evolved genotype of those males). What is less clear is why the manipulation of sexual selection had a larger

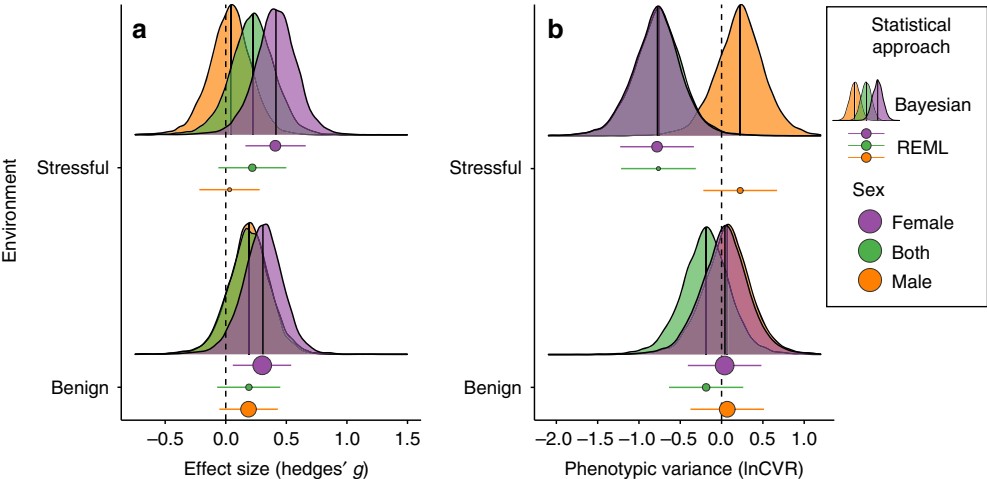

**Fig. 2** The roles of environmental stress and sex. **a** Sexual selection tends to increase the population mean values of fitness traits, especially for female traits and for populations living under stressful conditions. **b** Under stressful conditions, sexual selection tends to reduce the phenotypic variance in fitness traits, especially for traits measured in females or mixed-sex individuals. The points with error bars show the mean effect sizes and their 95% confidence intervals (CIs), determined from a meta-regression fit using restricted maximum likelihood (REML); the point sizes are proportional to the number of effect sizes (see Supplementary Tables 14 and 18). Results from Bayesian meta-regression are shown as posterior prediction density curves, with vertical lines indicating the median. Data used for generating this figure are available in the Source Data file

effect on female trait means and variances as opposed to males—this result is arguably the opposite of what one would expect, since it is males that experience stronger sexual selection. Below we discuss possible explanations for this result, in light of the core principle that the extent of adaptation depends on additive genetic (co)variance and the strength of selection[45,46].

First, it is possible that female traits show more additive genetic variation than male traits, causing female traits to respond more strongly to a change in selection. This hypothesis is plausible because males frequently do experience stronger selection than females[47], and sustained strong selection reduces heritability. A systematic review found no overall difference in mean heritability between male and female traits[48], but did record numerous instances in which trait heritability was higher for females than males[48]. The sex chromosomes provide another reason for sex-specific heritability. Males are heterogametic in most of the species in our sample (i.e. the species with XY or XO sex determination), which can reduce father-to-son heritability relative to mother-to-daughter, since sons do not inherit the larger sex chromosome from their fathers[14,49–51], potentially slowing the adaptation of male traits[51,52].

Second, selection on males might be weaker than selection on females, resulting in slower adaptation following the experimental manipulation of sexual selection. This explanation may initially seem implausible, because net selection on males is often stronger than on females[47], due in part to the elevated importance of sexual selection in males as opposed to females[53–56]. However, an oft-overlooked aspect is that selection might frequently be softer on males and harder on females[57], because the local competitive environment is usually more important for males than it is for females. For instance, a mediocre male genotype can have high fitness provided it outcompetes its local rivals, while low-fitness female genotypes might produce few offspring even when competing with other low-fitness females. Therefore, improvements in genetic quality might have stronger diminishing returns in males, possibly contributing to our finding that the genetic consequences of sexual selection lead to greater fitness benefits for females. Though this argument is speculative, we note that many experimental evolution designs exaggerate the sex difference in

the softness of selection, relative to expectations for large, natural populations[58–60]. For example, many studies[61–65] have evolved insects in small sub-populations, each containing one female and multiple males, whose progeny are then mixed and randomly sampled to create the next generation; this design ensures that successful males simply needed to outcompete their rival(s) in the same sub-population (soft selection), while each female's reproductive output is measured against the entire female population (hard selection).

Our results suggest that the greater benefit of sexual selection to females than males is magnified in stressful environments. Recent work has emphasised that environmental stress should reduce the strength of sexually antagonistic selection relative to selection that is concordant between sexes. Theoretical models reaching this conclusion[30,66] have been supported by some empirical work[12,31]; for example, one study found that high fitness males produced low fitness daughters under benign conditions, but high fitness daughters under stress[31]. However, other quantitative genetic studies have shown that stressful conditions do not always reduce sexual antagonism[24,27]. Variation in effects of sexual selection in stressful environments may be due to potentially variable responses amongst taxa[28] and environments. Notably, Connallon and Hall[30] predict that the dynamics of environmental change alter the strength of sexual antagonism; for instance, gradual directional selection may facilitate indefinite sexual antagonism, while rapid cyclical change can swiftly remove it. Our meta-analysis suggests that under directional selection imposed by environmental stress, sexual antagonism is likely dampened, allowing sexual selection to facilitate adaptation and persistence.

Although our meta-analysis revealed an overall positive effect of sexual selection, the variation in effect size across the dataset is high, as is often the case for studies in ecology and evolution[32]. Most of the heterogeneity was between studies (potentially due to differences between study designs and populations), while the taxon, number of generations of evolution and use of blinding had less impact on effect size. Experimental evolution studies cover relatively few taxa, and most focus on easy-to-culture invertebrates with similar mating systems and sex determination.

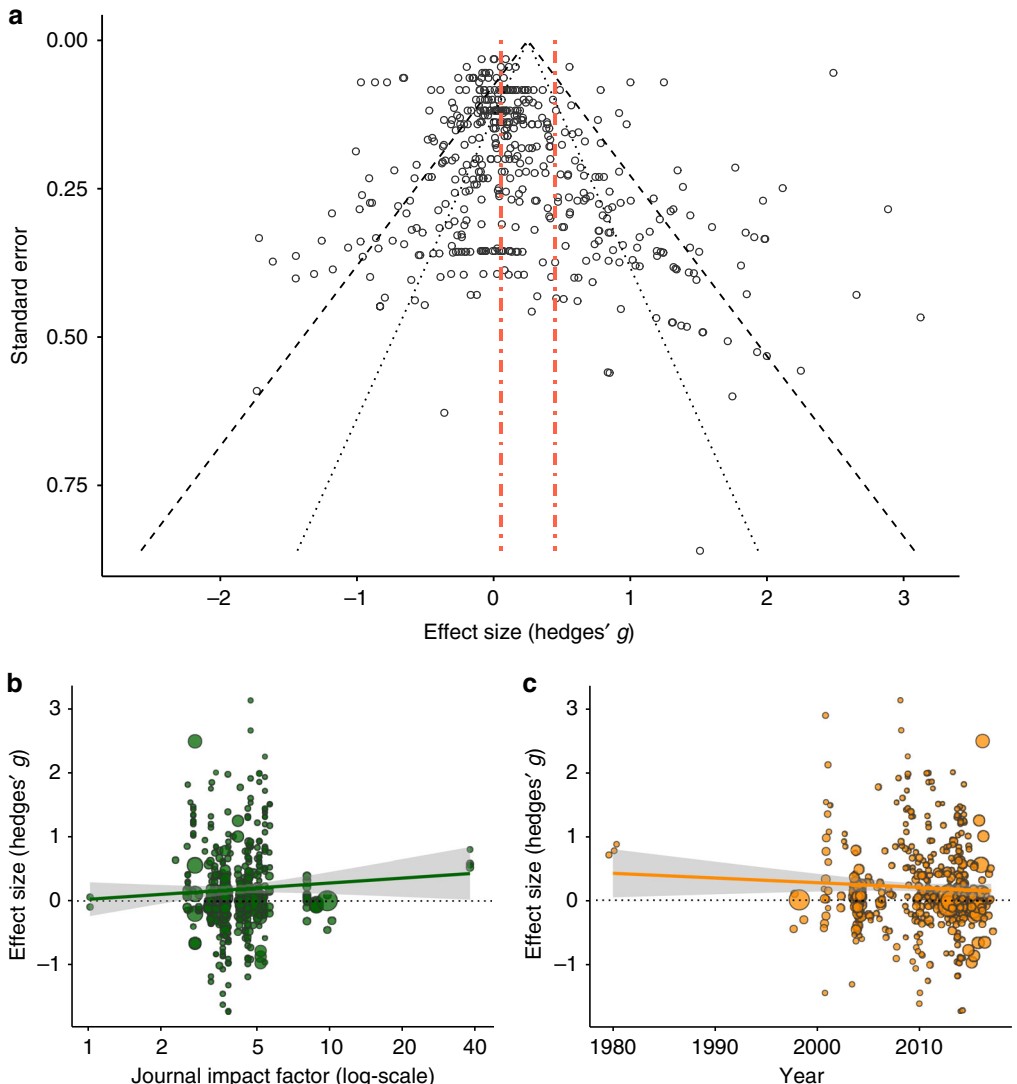

**Fig. 3** Tests for publication bias in the dataset. Tests of publication bias are mixed and suggest publication bias may be present. **a** Inspection and statistical tests of the funnel plot reveal large amounts of heterogeneity in the dataset with asymmetry from increased low-powered, large effect studies. **b** No significant correlation exists between journal impact factor and effect size. **c** additionally there is no significant correlation between effect size and year of publication when testing for the time-lag bias. For **b**, **c**, point size is proportional to the precision of the effect size (i.e. the inverse of its variance). For **a**, the dashed red lines represent the 95% confidence intervals (CIs) of the grand mean estimate for all effect sizes, and the black dotted and dashed lines depict the 95 and 99.8% CIs for the dataset. The grey envelopes in **b**, **c** represent the 95% CI of the linear regression. Data used for generating this figure are available in the Source Data file

However, a meta-analysis of macroevolutionary studies on sexual selection and speciation rate found no significant taxon-based differences across a diverse sample of vertebrates and invertebrates (fish, insects, birds, spiders, reptiles, mammals)[67], perhaps suggesting that our results would generalise to other taxa. On the other hand, biological differences between taxa could change the relationship between sexual selection and population fitness. For example, species where males and females have radically different morphology might have a reduced inter-sexual genetic correlation for fitness, such that sexual selection has fewer pleiotropic benefits for females, while for species with sexually selected male parental care, sexual selection might help the population by conferring high fitness to caring fathers.

Our findings have implications for fundamental and applied research. For example, the beneficial population-level consequences of sexual selection have been proposed as one possible resolution to the long-standing evolutionary puzzle regarding sexual reproduction[68]. If sufficiently strong, these benefits can more than compensate for the costs of sexual reproduction, and prevent sexual populations from being outcompeted by asexual mutants[9,10]. Sexual selection is also important for conservation[5] and captive breeding programs[69]. Within captive breeding programs, genetic diversity is often managed through the enforced monogamy of a strategically selected (genetically diverse) breeding pair[69]. Captive breeding programs may benefit from allowing sexual selection of 'good genes' or more compatible genes[70], or by increasing maternal investment by females paired with attractive males[71–73]. Additionally, our findings imply that anthropogenic environmental changes that reduce the opportunity for sexual selection, such as eutrophication, pesticides, artificial light and noise pollution, could reduce the genetic quality of the population, and potentially compromise its long-term persistence[74–77]. Equally, our results support recent evidence that human activities that directly counteract sexual selection, such as selective

harvesting of the largest or most ornamented males, can lower population fitness[78]. Based on the weight of evidence from experimental evolution, we suggest that sexually selected populations may be more resilient to environmental change, including anthropogenic environmental pressures, over relevant time scales.

## Methods

**Literature search**. We searched ISI Web of Science and Scopus on 9 June 2017 for peer-reviewed, English language studies that manipulated the presence or strength of sexual selection using experimental evolution, and then measured some proxy of population fitness. A detailed list of search terms is given in the Supplementary Information (Supplementary Methods).

After removing duplicates, we read the titles and abstracts of the remaining 1015 papers and removed those that did not fit our inclusion criteria (typically because they did not present primary experimental evolution data). This left 130 papers, for which we read the full text and applied the inclusion criteria outlined in the PRISMA (Preferred Reporting Items for Systematic Reviews and Meta-Analyses) diagram (Fig. 4). Briefly, we included studies that (1) were conducted in a dioecious animal, (2) experimentally manipulated the strength of sexual selection (e.g. via experimentally enforced random monogamy or an altered sex ratio) for at least one generation and (3) measured a trait that we judged to be a potential correlate of population fitness. This third criterion is the most subjective, because there is rarely enough data to determine whether a particular trait is (or is not) correlated with population fitness. We therefore relied on our best judgement when deciding what outcomes were correlated with population fitness. We categorised the fitness outcomes into three categories: ambiguous, indirect and direct (detailed in Supplementary Table 1). Briefly, ambiguous measures of fitness were those that are reported to have an unclear or variable association with fitness (e.g. body size, mating duration, early fecundity and male reproductive success). Indirect fitness components were those that are often used as a proxy of fitness, but do not directly measure aspects of success in reproduction or population viability (e.g. lifespan, mating success and ejaculate quality/production). Finally, direct measures of fitness (female/mixed sex reproductive success, offspring viability and extinction rate) are those that measure fitness through components of reproduction or long-term viability. The Supplementary Methods describe why each of the 130 papers was included or excluded (Supplementary Table 2).

Of these 130 papers, 62 were excluded based on the PRISMA criteria (Fig. 4). Additionally, three papers presented insufficient information to calculate effect size. In these cases, we contacted the authors and attempted to obtain the missing data, with partial success. The final meta-analysis included data from 65 papers.

**Data extraction**. From each paper, we first attempted to extract the arithmetic means, standard deviations and sample sizes of each of the different treatment groups, which facilitate calculation of effect size (see below). Typically, there were two or three treatments, which varied in the strength of sexual selection on males through manipulations to the adult sex ratio; in these cases we considered treatments with the greater male-to-female ratio to be the high sexual selection treatment group. For some papers, summary statistics were not written down, but were presented in a figure such as a bar chart: in these cases, we extracted the data using *WebPlotDigitizer* v.3.12[79]. If the treatment means were not reported (and the raw data were unavailable), we instead calculated effect size from test statistics comparing treatment means (e.g. *F*, *t*, *z* or $\chi^2$ values), which we used to estimate effect size using several formulae (see below).

Where possible, we extracted data for each independent replicate or experimental evolution line within a study; otherwise, we used pooled treatment means. For studies that repeatedly measured the same population across multiple generations, we only extracted data for the last reported generation.

In addition to the data used to calculate effect size, we collected a set of moderator variables for each paper (see the Source Data file and associated Supplementary Information). The moderators were selected due to their ready availability, and because we hypothesised that they might explain some of the observed heterogeneity in effect size. A key moderator was whether the environmental conditions that a population evolved under were stressful (e.g. elevated mutation load, novel/sub-optimal food source, increased sub-lethal temperatures). Additionally, we collected details for each effect size on: sex (male, female or a mixed sample of both), taxon (flies, beetles, mice, nematodes, mites, crickets and guppies), the presence/absence of blind methodology and number of generations a treatment group underwent experimental evolution. In the interests of creating a useful data resource, we also recorded details about each experiment that were not formally analysed due to a shortage of data, such as the type of sexual selection that was manipulated (pre-copulatory, post-copulatory or both) and the male-to-female ratio, which is included in the Source Data file.

**Effect size calculation**. For each measurement of each pair of treatments, we estimated the standardised effect size Hedges' *g*[80]. Similar to Cohen's *d*, Hedges' *g* expresses the difference in means in terms of standard deviations (making it dimensionless), but it is more robust to unequal sampling and small sample sizes[81]. For comparisons of extracted treatment means, we calculated Hedges' *g* using the

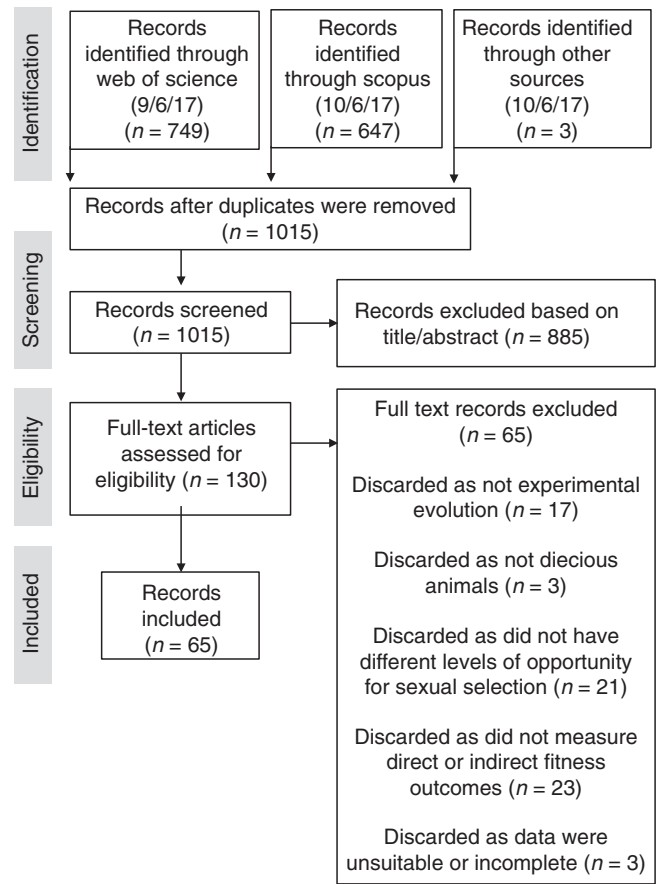

**Fig. 4** PRISMA diagram. Flow of inclusion and exclusion of studies identified during the literature search, presented as a PRISMA diagram with number of published papers in brackets

mes function in the *compute.es* R package[82]. To calculate Hedges' *g* from test statistics, we used the fes, chies and tes functions in the *compute.es* package (for *F*, $\chi^2$ and *t* statistics, respectively). The propes function was used to calculate effect size from a difference in proportions; in two cases[83,84], a proportion was equal to one (producing infinite effect sizes), and so we subtracted one from the numerator when estimating Hedges' *g*. In all cases, we selected a direction for the effect size calculation such that in our meta-analysis, negative effect sizes indicate that the removal of sexual selection was associated with higher fitness trait values, and positive effect sizes indicate higher fitness when sexual selection was elevated or left intact. We also inverted the sign of effect sizes pertaining to measurements that are expected to be negatively related to population fitness (e.g. parasite load, mutation load, extinction risk/rate, mating latency (males) and rate of senescence). Because many of our 65 papers measured multiple fitness outcomes, studied multiple replicate populations or had three or more sexual selection treatments, we calculated a total of 459 effect sizes.

Additionally, using studies that presented means, standard deviations and sample sizes (*n* = 352) we were able to calculate an alternative measure of effect size: the *lnRR*[85,86]. The *lnRR* was used as a supplement to Hedges' *g* because it relaxes the assumption in equal variances between control and treatment groups (homoscedasticity).

For the meta-analysis testing whether sexual selection affects phenotypic variance (as opposed to the mean), we estimated the difference in variance between each pair of treatments using the natural logarithm of the ratio between the coefficient of variation for each group (termed *lnCVR*[34]: ln (CVfitness$_{SS\ high}$/CVfitness$_{SS\ low}$). The use of *lnCVR* allows us to determine the effects of sexual selection on phenotypic variance, with the coefficient of variation implicitly controlling for the mean-variance relationship seen in the dataset (Supplementary Fig. 5). As a supplement, we also calculated the natural logarithm of the absolute ratio between the absolute variation for each group (*lnVR*) in order to assess the impact of sexual selection on trait variance, irrespective of their magnitudes[34]. The calculation of *lnCVR* and *lnVR* relies on the availability of arithmetic means, standard deviations and sample sizes for the two treatment groups[34,87], and so we were only able to calculate *lnCVR* and *lnVR* for 354 of 459 comparisons.

**Mixed-effects meta-analysis.** First, we obtained a weighted mean effect size (Hedges' *g*) for the entire dataset, using both Bayesian and REML approaches for completeness. The weighted mean was obtained by fitting a model with no moderator variables (i.e. fixed effects), but fitness component (e.g. body size, female reproductive success), study ID and taxon as random/group-level effects. That is, we separately model correlations between different effect sizes sourced from the same study, taxon or pertaining to the same fitness component, and account for these interdependencies when estimating the overall effect. Given the small number of phylogenetically diverse species, we did not utilise phylogenetic corrections within the models. In our meta-analyses, we report Bayes factors (BF), giving the likelihood ratio that the focal effect size differs from zero $BF_{>0}$.

Second, we fixed the relationship to fitness class (Ambiguous, Indirect or Direct) as a moderator variable in Bayesian and REML models (whilst maintaining study and taxon as group-level effects) to derive predictions for effect size within each of the three fitness-relationship classes, using the relevant `predict` functions for each of the R packages used (see below). This meta-analysis was then supplemented by another model where we fixed fitness component as a moderator variable (e.g. immunity, lifespan, offspring viability and female reproductive success); predictions for this model on the 22 fitness components were derived as above. Alternatively, to assess the impact of sexual selection on each fitness component independently of one another, we conducted separate meta-analyses ($n = 18$); subset for each fitness trait with more than three effect sizes. These models were were intercept only REML models with study and taxon as group-level effects. Further details on model parameters can be found by accessing the R code.

Third, we measured the impact of environment, sex and their interaction on the effect size (Hedges' *g*, lnRR, lnCVR and lnVR) associated with the manipulation of sexual selection, by fitting these predictors as moderators in a pair of separate mixed-effects meta-analyses. These meta-analyses were restricted to effect sizes calculated from unambiguous outcomes (i.e. those scored as being directly or indirectly related to population fitness), as well as those where we were able to define the environmental conditions as either stressful or benign (Hedges' *g*: $n = 330$; lnRR, lnCVR and lnVR: $n = 269$). We again fit study ID, fitness component and taxon as random/group-level effects. Models investigating other moderators such as number of generations and blinding are presented in Supplementary Table 9.

For our meta-analyses investigating the effects of environment and sex on the magnitude and variance of fitness-related traits, we provide estimates of heterogenity present in the dataset. We use the statistic $I^2$ as an estimate of the proportion of variance in effect size that is due to differences between levels of a random effect (e.g. studies)[88]. $I^2$ is preferred over other statistics as it is independent of sample size, is easily interpretable and can be partitioned between random effects[32]. Within ecology and evolution heterogeneity in datasets is often high, with the mean $I^2$ from 86 studies above 90%[33].

Meta-analyses fit by REML were implemented in the *metafor* R package[89], while their Bayesian equivalents used the R package *brms* to run models in Stan[90].

**Publication bias.** We tested for publication bias via funnel plots, using Egger's test to quantify plot asymmetry[91]. Additionally, we tested for time-lag bias[36], in which effect size magnitudes decline over time as more data are collected. Additionally, we assessed a potential source of publication bias through the correlation between effect size and journal impact factor[37], which can arise if null or countervailing results are more difficult to publish (impact factors were from *InCites Journal Citation Reports*).

**Reporting summary.** Further information on research design is available in the Nature Research Reporting Summary linked to this article.

## Data availability

All data are freely available on Github (https://github.com/JustinCally/SexualSelection). Additionally, the source data used in this meta-analysis is provided as a Source Data file alongside this manuscript.

## Code availability

The code used to perform all analyses is presented as an annotated HTML report at https://justincally.github.io/SexualSelection/, and the raw R Markdown files are archived at https://github.com/JustinCally/SexualSelection.

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

## Acknowledgements

We are very grateful to Paco Garcia-Gonzalez for sharing a collection of literature that helped to guide our study, and to Tim Connallon for comments on the manuscript.

## Author contributions

J.G.C., D.S.-F. and L.H. designed the study; J.G.C. collected the data; J.G.C. and L.H. conducted the meta-analyses; J.G.C. wrote the first draft of the manuscript; J.G.C., D.S.-F. and L.H. contributed substantially to all further revisions.

## Additional information

**Competing interests:** The authors declare no competing interests.

