## [Peer Review File · Nature Communications]

Reviewers' Comments:

Reviewer #1:

Remarks to the Author:

The manuscript presents results of a meta-analysis assessing the effect of sexual selection on population fitness. This topic has been increasingly studied over the past decade or so, and the time seems ripe for a synthesis. The authors have done good job searching the literature – it includes all relevant papers I could remember, and the results should be of interest to a broad range of evolutionary biologists. I cannot expertly judge on state-of-the-art methodology, but I do have several general concerns about how the analyses were performed.

Firstly, it seems that several male traits like to be under sexual selection, but not related to population fitness were included. I don't think this is correct given the question being asked (i.e. whether sexual selection increases population fitness). I think traits which are directly under sexual selection (eg. attractiveness, reproductive success) should not be pooled with traits which may respond to manipulation of sexual selection indirectly (eg. male development time, survival) and can affect population fitness. Distinguishing between both types of trait could actually be revealing – e.g. could expose trade-offs between sexually selected traits and fitness components unrelated to reproductive competition (see eg. Radwan et al. 2015 *Evol Biol*), and thus potentially explain lower effect of sexual selection on (pooled) male traits than on female traits.

Secondly, looking at Figure 1, one notices apparent anomalies, for example significantly positive slope for male attractiveness, based on studies with average effect size close to zero. I think this (and few others) surprising estimates may result from fitting random terms across all trait types (fitted as a fixed factor, second model); I guess fitting interaction (random slopes) would not be feasible for some categories including few data points, but some of them could easily be pooled in wider categories.

Thirdly, type of trait measured explained 35% variance, but the authors do not explore this any further. However, examination of Fig. 1 suggest that some indirect/ambiguous fitness measures account for much of this heterogeneity and they generally have higher average effect sizes than direct measures (except for immunity). I'd like to see if the authors recover their main result if they only direct measures.

Another major problem I have with the manuscript concerns interpretation of the very intriguing finding that the response to manipulation of sexual selection was stronger for female traits compared to male traits. I'm confused by the authors' explanation: do they assume sexual selection acted directly on females, and not only indirectly, via males? Only then things like mother to daughter heritability, or hard selection on females, should matter.

Perhaps the effect on females is indeed direct, and results from stress imposed by polygamous treatment, which magnifies direct, hard selection on females? This would be an important finding, and perhaps the authors could test it with their dataset by contrasting middle-class-neighborhood-like studies from those which allowed for female evolution. But if correct, this explanation is not exactly the effect of sexual selection, but rather enhanced selection of females due to enhanced (male induced) stress, so the interpretation of results should change.

Other comments:

l. 30 – reviews on sexual conflict are OK to cite here, but there are empirical papers actually demonstrating correlation between male sexual selected traits (Harano et al. 2011; Plesnar et al. 2014) which should also be cited.

I. 151 – the authors discuss beneficial effects of sexual selection on direct fitness measures such as reproductive success or offspring viability, but estimates for both of these measures actually overlapped zero! Perhaps joint analysis of direct fitness measures, as I suggested above, could support this conclusion, but currently this is an overstatement.

In the discussion the authors say they included the number of experimental evolution generations, but I could not find this information in methods.

Fig. S1 is not referred to in the main texts, is it different from Fig. 2, except that the latter contains predicted average values for fitness components?

Jacek Radwan

Reviewer #2:

Remarks to the Author:

This meta-analysis investigated the consequences of sexual selection experiments on trait mean and variance (comparing sexually selected groups vs control groups). Overall, the authors observe sexual selection usually increase mean and reduce variance (especially in females). The authors conclude sexual selection's benefits outweigh its detrimental effects. This meta-analysis is extremely well conducted (the use of both likelihood-based and Bayesian models for robustness), and although I am not an expert on this topic, I really enjoyed reading it, and it was very clear. Especially, I am impressed with the detailed supplement which showed the code and analysis. However, I have several comments regarding their analysis, which will increase the robustness of results and thus conclusions.

1 --- The use of Hedges' g . I understand that Hedges' g was probably used because it can take interval measurements as well as ratio measurements (InCVR can only take ratio measurements). However, g cannot really deal well with heterogeneity between two groups (i.e. experimental and control groups having different variances). This is why Hedges' g (Cohen's d) was criticized earlier.

Osenberg, C. W., O. Sarnelle, and S. D. Cooper. 1997. Effect size in ecological experiments: the application of biological models in meta-analysis. *American Naturalist* 150:798-812.

As a response, they come up with log response ratio (lnRR) - see

Hedges, L. V., J. Gurevitch, and P. S. Curtis. 1999. The meta-analysis of response ratios in experimental ecology. *Ecology* 80:1150-1156.

I recommend that the authors use lnRR for their effect size for mean comparison as well as Hedge's g to see the robustness of their conclusions for the mean.

2 --- The authors may consider also doing another set of meta-analyses using lnVR (log variability ratio - proposed in Nakagawa et al. 2015).

Nakagawa, S., R. Poulin, K. Mengersen, K. Reinhold, L. Engqvist, M. Lagisz, and A. M. Senior. 2015. Meta-analysis of variation: ecological and evolutionary applications and beyond. *Methods in Ecology and Evolution* 6:143-152.

As one can see in Figure 2, the mean results (Hedges' g) are a mirror image of the variance results (this makes sense CV controls for means). I would like to see what the absolute change in variances.

Probably the authors can put the analysis using InVR in the supplement. The results of this analysis can be discussed. Also, the mean-variance relationship between mean and variance (sd) should be verified (e.g. plot log mean and log sd or log variance).

3 --- I^2 needs to be explained. I^2 is proposed originally here:

Higgins, J., and S. Thompson. 2002. Quantifying heterogeneity in a meta-analysis. *Statistics in Medicine* 21:1539 - 1558.

But later expanded in here for mixed models (hierarchical models)

Nakagawa, S., and E. S. A. Santos. 2012. Methodological issues and advances in biological meta-analysis. *Evolutionary Ecology* 26:1253-1274.

This is what the authors use. Also, it will be good to put the degree in I^2 in the context. To do this see this paper:

Senior, A. M., C. E. Grueber, T. Kamiya, M. Lagisz, K. O'Dwyer, E. S. A. Santos, and S. Nakagawa. 2016. Heterogeneity in ecological and evolutionary meta-analyses: its magnitude and implications. *Ecology* 97:3293-3299.

4 --- Publication tests have been conducted on, I think, "meta-analytic" residuals results as suggested in Nakagawa and Santos 2012. One could use such residuals to conduct the trim and fill method and see how much the mean could move (see Nakagawa and Santos 2012). One should remember the funnel asymmetry could be caused by the presence of heterogeneity. See:

Egger, M., G. Smith, M. Schneider, and C. Minder. 1997. Bias in metaanalysis detected by a simple, graphical test. *Br Med J* 315:629 - 634.

5 --- Figure 3 - are these grey envelopes 95% CI?

6 --- the title - do the authors include "a systematic review" - so "a systematic review and meta-analysis" - these two things are different - see:

Nakagawa, S., and R. Poulin. 2012. Meta-analytic insights into evolutionary ecology: an introduction and synthesis. *Evolutionary Ecology* 26:1085-1099.

Hope my comments are useful.

Reviewer #3:

Remarks to the Author:

Sexual selection improves population fitness: a meta-analysis

This manuscript addresses the question of whether, on average, sexual selection has a net beneficial or detrimental effect on population fitness, using a meta-analysis of experimental evolution studies comparing the fitness of populations under different intensities of sexual selection. The analyses consider the effects of sexual selection on both the mean and variance of population fitness, and test whether these effects differ for stressful vs. benign environments, or depend on the measure of population fitness.

These are issues of longstanding interest, and as there is now quite a wealth of experimental evolution studies addressing these questions a meta-analysis synthesising their findings is timely.

The results of this meta-analysis broadly concur with current theoretical predictions, and are likely to interest a wide readership. Across all studies, environment types and fitness measures there was a small positive effect of sexual selection on mean population fitness, although effect sizes varied with the fitness measure used. The positive effect of sexual selection on fitness was strongest for females in stressful environments, who also showed reduced variance in fitness under sexual selection.

The authors discuss these findings thoughtfully, although they focus more on the (weaker) effects found for 'direct' measures of fitness while somewhat neglecting the effects seen for 'indirect' fitness measures (e.g. of attractiveness, mating latency, lifespan, ejaculate traits). These potentially deserve more consideration, especially as many appear to be male-limited traits or measured mainly in males, that would appear to have a direct bearing on male mating success (if an indirect link to overall fitness), yet despite this the overall effect of sexual selection on male fitness is not significant.

I have only a few more specific comments:

Does your approach consider variation in the intensity of sexual selection exclusively on males? This is worth noting (e.g. statements such as that on line 98-99 might be amended to "Sexual selection on males significantly improved female fitness.")

Experiments manipulating the sex ratio to alter sexual selection might simultaneously decrease sexual selection on males while increasing sexual selection on females. Would this be classified as reduced sexual selection in your analyses? I don't think you included this aspect of study design as a moderator variable in any of your analyses – given that close to half of the effects you include come from "alternative manipulations" (line 70) perhaps you could test whether this affects effect size? It seems possible that sexual selection on females, and not only on males, could affect population fitness – potentially even more directly than sexual selection on males. And might differing extents of sexual selection in each sex interact with the differences you saw in fitness measured in males vs. females?

L111-118 It seems a little odd that there is no residual heterogeneity in your I2 estimates. From the supplemental information it is not entirely clear to me how you have adapted the function that is under development in "metaAidR", but can you double check that you have appropriately incorporated the residual variance into these estimates?

L307-311 Please clarify here that $\ln CVR$ was calculated as the ratio $\ln(CVfitness[SS]/CVfitness[no SS])$

Fig. 3 Please explain the dashed red and black lines in the figure caption.

Tables 1, 2 Please check – the parameter estimates for 'Female sex' and test for 'Female > Male' are bolded inconsistently between the REML and Bayesian models where these estimates are identical.

Comments from Reviewer 1 (Prof. Jacek Radwan)

The manuscript presents results of a meta-analysis assessing the effect of sexual selection on population fitness. This topic has been increasingly studied over the past decade or so, and the time seems ripe for a synthesis. The authors have done good job searching the literature – it includes all relevant papers I could remember, and the results should be of interest to a broad range of evolutionary biologists. I cannot expertly judge on state-of-the-art methodology, but I do have several general concerns about how the analyses were performed.

Many thanks for your time and for you very helpful feedback.

Firstly, it seems that several male traits like to be under sexual selection, but not related to population fitness were included. I don't think this is correct given the question being asked (i.e. whether sexual selection increases population fitness). I think traits which are directly under sexual selection (eg. attractiveness, reproductive success) should not be pooled with traits which may respond to manipulation of sexual selection indirectly (eg. male development time, survival) and can affect population fitness. Distinguishing between both types of trait could actually be revealing – e.g. could expose trade-offs between sexually selected traits and fitness components unrelated to reproductive competition (see eg. Radwan et al. 2015 Evol Biol), a thus potentially explain lower effect of sexual selection on (pooled) male traits than on female traits.

We appreciate the reviewer's point that distinguishing between traits that are direct targets of sexual selection, versus those that respond more indirectly, could be revealing. For this reason, we present separate meta-analytical measurements for the effects of sexual selection on each of the various fitness components in the supplementary material. This enables the reader to gauge the effect of sexual selection each of the fitness components, e.g. to test whether direct targets of sexual selection (such as attractiveness) respond more strongly to sexual selection than traits such as immunity or longevity with a more indirect link to reproduction.

However, because the subject of our study is population fitness, we classified fitness components according to their relationship to population fitness, rather than their relationship with sexual selection. We took this decision because the relationship between sexual selection and population fitness is contentious, while there is no controversy over how sexual selection should affect traits like attractiveness (almost by definition, individuals born under sexual selection should be more attractive whenever attractiveness is hereditary).

In the revision, we have slightly altered which traits were classified as having a “direct”, “indirect”, or “ambiguous” relationship with population fitness, in light of the reviewer's feedback (see Table S1 for a

breakdown of how all the traits were classified). Because we think that attractiveness is only tenuously linked to population fitness, and we have re-classified its relationship with population fitness as “ambiguous” (as opposed to “indirect” in the original manuscript). Additionally, we have split the trait “Reproductive success” by sex, in light of the fact that male and female fitness are maximised in different ways and are likely to have differentially related to population fitness. The revised manuscript and its main figures show that this reclassification had little impact on our main conclusions.

Secondly, looking at Figure 1, one notices apparent anomalies, for example significantly positive slope for male attractiveness, based on studies with average effect size close to zero. I think this (and few others) surprising estimates may result from fitting random terms across all trait types (fitted as a fixed factor, second model); I guess fitting interaction (random slopes) would not be feasible for some categories including few data points, but some of them could easily be pooled in wider categories.

For reference, we here present the result for male attractiveness that the reviewer mentioned. The table shows the estimate from the meta-analysis, and the figure shows the $n = 6$ effect sizes that underlie it, with the mean effect size plotted as vertical lines.

Table 1: Model predictions for Male Attractiveness

Fitness Component	Bayes Prediction	Bayes SE	Bayes LCI	Bayes UCI	n	BF	REML Prediction	REML SE	REML LCI	REML UCI
Male Attractiveness	0.302	0.14	0.031	0.59	6	5.8e+01	0.298	0.111	0.081	0.515

We agree that alone, these 6 effect sizes do not provide strong evidence that the true mean effect size is positive. But given that 4/6 of the effect sizes are positive, and these were all either precisely measured or quite large, the model’s conclusion that there is very weak evidence for a positive effect seems correct.

Additionally, regarding the comment about random slopes, we were not sure what revised model the reviewer is proposing. To clarify things, we first note that the current model has the following attributes:

- *Effect size* (i.e. the effect of sexual selection treatment on some fitness component) as the response variable, weighted by the associated standard error
- The *type of fitness trait* (e.g. fecundity, lifespan, attractiveness) as a fixed factor or ‘moderator variable’
- The *Study ID* as a random intercept (to model the similarity among effect sizes derived from the same study)
- The *Taxon* as a random intercept (to model the similarity among effect sizes pertaining to the same taxon)

Given this, we suppose that the reviewer was proposing to add the fixed factor *type of fitness trait* as a random slope to one or both of the two random intercept terms. This would allow each fitness trait to show a different amount of inter-study and inter-taxon variation. We suspect that such a model would be overly-complex given our dataset, and it is unlikely to provide a statistically significantly better fit than the more parsimonious model that we currently use. Rather few studies in our dataset measured multiple traits, and there are several taxa in which only one or two trait types have been measured, meaning that the model would not be able to accurately measure the variance explained by the proposed random slopes.

Alternatively, it is possible that the reviewer meant for us to build a model that allows the response to sexual selection treatment to differ among the fitness traits. In pseudocode, this model would look something like `Trait_value ~ Sexual_selection_treatment * Type_of_trait`. However, this would represent a misunderstanding: we did not analyse the raw trait values from the original studies, but rather we collected their effect sizes and combined them to estimate the average effect size for each of the different fitness traits. Our present analysis already allows for the possibility that the response to sexual selection is different for different types of fitness trait. Apologies in advance if we have missed the point you were making.

In light of this comment, we have changed our main figure for greater clarity and simplicity (we now group the traits by their relationship to population fitness, as opposed to the numerous individual types of types). Specifically we have changed Figure 1 to a simpler style of forest plot, with the big forest plot that was formerly Figure 1 now moved to the Supplementary Material (Fig. S1).

Thirdly, type of trait measured explained 35% variance, but the authors do not explore this any further. However, examination of Fig. 1 suggest that some indirect/ambiguous fitness measures account for much of this heterogeneity and they generally have higher average effect sizes than direct measures (except for immunity). I'd like to see if the authors recover their main result if they only direct measures.

We followed this suggestion and re-ran the model on ‘direct only’ and compared this to our previous results (both are presented below for comparison). The main result remains the same: the positive effect of sexual selection on fitness is magnified for females in stressful environments. However, two things have changed in terms of statistical significance.

Firstly, for the ‘direct only’ model the predicted effect size for males in stressful environments is now significantly negative (*Hedges’ g* = -0.45; 95 % CIs = -0.70 to -0.19) as opposed to showing no difference. However, there are only two effect sizes for males in stressful environments using the ‘direct only’ subset of the data, so this is not very strong evidence that the direct and indirect measures differ considerably.

Secondly, the predicted effect size for females in benign environments in the ‘direct only’ model are non-significantly positive, unlike the “direct and indirect” model where this result was statistically significant.

We now include the results recovered of the ‘direct only’ model in the supplementary material, and mention them in the main manuscript. We have left the ‘direct plus indirect’ model as our main focus, because its sample size and thus statistical power is much higher than the ‘direct only’ model ($n = 289$ vs $n = 159$ effect sizes).

N.B. Using the corrected I^2 formula (see reviewer 3 comments) the type of trait measured only accounts for ~ 0.5 % of the total heterogeneity (for *Hedges’ g*) and thus we have not added further discussion.

Table 2: Model predictions using only direct measures of fitness

Sex	Environment	Prediction	SE	CI.lb	CI.ub	n
Male	Benign	0.131	0.12	-0.098	0.36	13
Both	Benign	0.104	0.12	-0.137	0.34	15
Female	Benign	0.091	0.11	-0.121	0.3	86
Male	Stressful	-0.45	0.13	-0.707	-0.19	2
Both	Stressful	0.135	0.12	-0.108	0.38	12
Female	Stressful	0.312	0.11	0.093	0.53	31

Another major problem I have with the manuscript concerns interpretation of the very intriguing finding that the response to manipulation of sexual selection was stronger for female traits compared to male traits. I'm confused by the authors' explanation: do they assume sexual selection acted directly on females, and not only indirectly, via males? Only then things like mother to daughter heritability, or hard selection on females, should matter.

We agree that this section could have been much clearer, and have made changes in light of this comment to clarify what we meant. We believe that experiments that manipulate sexual selection on males will alter female fitness traits, even in species where females experience no sexual selection, because of mechanisms such as intra- and inter-locus sexual conflict. That is, evolution in males will almost always have pleiotropic consequences for females, or change how males affect females (i.e. sexual selection on males causes evolution of males' indirect genetic effects on females).

Consider for example the classic Holland and Rice-style experiment, where one removes sexual selection by enforcing random monogamy in a species with 'traditional sex roles', such as *Drosophila*. Males are predicted to evolve to be less harmful to females in these experiments, and they are also predicted to evolve a more female-like overall phenotype, due to the removal of selection on male-specific functions involved in sexual selection. Both of these male adaptations are expected to cause a genetically-based change in female traits: the former as a result of inter-locus coevolution (e.g. female resistance should evolve in response to the reduction in male-induced harm), and the latter as a result of pleiotropy between the sexes. We thus might

predict that females would adapt even more than males if female traits have more genetic variation, or if selection is ‘harder’ on females than males. These explanations are speculative and *post hoc*, and so in the revision we are careful to identify them as such. Our result is arguably opposite to what one would predict, because naively one would expect sexual selection to have greater effects on the phenotype of males, and we now spell this out as well.

Perhaps the effect on females is indeed direct, and results from stress imposed by polygamous treatment, which magnifies direct, hard selection on females? This would be an important finding, and perhaps the authors could test it with their dataset by contrasting middle-class-neighborhood-like studies from those which allowed for female evolution. But if correct, this explanation is not exactly the effect of sexual selection, but rather enhanced selection of females due to enhanced (male induced) stress, so the interpretation of results should change.

Unfortunately, too few studies have manipulated sexual selection (independently of *all* selection) using the middle class neighborhood design for a reliable test. However we did collect data on how sexual selection was manipulated: many studies imposed random monogamy, but other studies instead manipulated the adult sex ratio (e.g. 3 males 1 female, 2 of each, or 3 females 1 male). In the revised Supplementary Material we now included “manipulation type” as a two-level moderator variable. We find no effect of manipulation type on effect size, suggesting that both ways of removing sexual selection had a similar effect on population fitness traits. This result can be found in the revised Table S10 (moderator labelled as Enforced MonogamyYES).

Other comments:

l. 30 – reviews on sexual conflict are OK to cite here, but there are empirical papers actually demonstrating correlation between male sexual selected traits (Harano et al. 2011; Plesnar et al. 2014) which should also be cited.

Thank you, we have added citations of those papers as suggested.

l. 151 – the authors discuss beneficial effects of sexual selection on direct fitness measures such as reproductive success or offspring viability, but estimates for both of these measures actually overlapped zero! Perhaps joint analysis of direct fitness measures, as I suggested above, could support this conclusion, but currently this is an overstatement.

Thank you for catching this – we have tempered our conclusions in the relevant section of the Discussion, since the results provide only moderate evidence for a positive effect.

In the discussion the authors say they included the number of experimental evolution generations, but I could not find this information in methods.

For relevant results, please see the section of the results that reads “Other moderator variables that we examined had minimal impacts on effect size (Figure S2, Table S10). Specifically, effect size did not depend on whether or not the study was conducted blind (Figure S6), nor on the number of generations for which the experimental evolution study was run (Figure S7, S8).” In the methods, we wrote “Additionally, we collected details for each effect size on: sex (male, female or a mixed sample of both), taxon (flies, beetles, mice, nematodes, mites, crickets and guppies), blinding of researchers to treatments and number of generations a treatment group underwent experimentally evolution.”

Fig. S1 is not referred to in the main texts, is it different from Fig. 2, except that the latter contains predicted average values for fitness components?

Our supplementary material contains all of our main figures (such as Figure 2), in order to show the data and the code that was used to generate them. We have changed the numbering of the supplementary figures, so that e.g. Figure 1 in the HTML supplement is correctly labelled as Figure 1 and not as Fig. S1, to make it clearer which are the same.

Comments from Reviewer 2

This meta-analysis investigated the consequences of sexual selection experiments on trait mean and variance (comparing sexually selected groups vs control groups). Overall, the authors observe sexual selection usually increase mean and reduce variance (especially in females). The authors conclude sexual selection's benefits outweigh its detrimental effects. This meta-analysis is extremely well conducted (the use of both likelihood-based and Bayesian models for robustness), and although I am not an expert on this topic, I really enjoyed reading it, and it was very clear. Especially, I am impressed with the detailed supplement which showed the code and analysis. However, I have several comments regarding their analysis, which will increase the robustness of results and thus conclusions.

Many thanks for your time and attention, and for the valuable feedback.

1 — The use of Hedges' g . I understand that Hedges' g was probably used because it can take interval measurements as well as ratio measurements ($\ln\text{CVR}$ can only take ratio measurements). However, g cannot really deal well with heterogeneity between two groups (i.e. experimental and control groups having different variances). This is why Hedges' g (Cohen's d) was criticized earlier.

Osenberg, C. W., O. Sarnelle, and S. D. Cooper. 1997. Effect size in ecological experiments: the application of biological models in meta-analysis. *American Naturalist* 150:798-812.

As a response, they come up with log response ratio ($\ln\text{RR}$) - see

Hedges, L. V., J. Gurevitch, and P. S. Curtis. 1999. The meta-analysis of response ratios in experimental ecology. *Ecology* 80:1150-1156.

I recommend that the authors use $\ln\text{RR}$ for their effect size for mean comparison as well as Hedges' g to see the robustness of their conclusions for the mean.

After reading those papers, we agree that Hedges' g is imperfect, and so we checked the robustness of our meta-analysis using two alternative measures of effect size. Firstly, we present the log response ratio ($\ln\text{RR}$) as suggested by this reviewer. Secondly, we present a modified version of Hedges' g that attempts to account for the heterogeneity issue (Bonett 2009), named SMDH (standardized mean difference with heteroscedastic population variances in the two groups). Here are the findings from our new analyses using $\ln\text{RR}$ and SMDH:

Firstly, we found that Hedges' g and SMDH are highly correlated (in means and variance). This is reassuring because it suggests that analyses based on g and SMDH are likely to yield the same results.

Secondly, we note that lnRR and SMDH can only be calculated in cases where the primary study reported means, standard deviations and sample sizes (352 out of 459 primary effect sizes). This is a limitation of lnRR and SMDH, as we must discard roughly 25% of the data in order to use them instead of Hedges' g . We therefore think it is best to focus primarily on Hedges' g despite its minor limitations, and to use lnRR and SMDH for sensitivity analysis in the online supplementary material.

Sex	Environment	Hedge's g sample size	lnRR sample size
Male	Benign	83	73
Both	Benign	15	12
Female	Benign	125	110
Male	Stressful	12	8
Both	Stressful	18	6
Female	Stressful	36	27
Total	-	289	236

Thirdly, if we conduct a meta-analysis using lnRR as the effect size we obtain very similar results with all our main conclusions unchanged, keeping in mind that lnRR and Hedges' g are on different scales (see the revised Supplementary Material). Specifically,

- The effect of sexual selection on population fitness tends to be positive.
- This effect is magnified for females in stressful environments with a significantly positive interaction term (stress*females).

Comparison between Hedge's g and $\ln RR$ effect sizes

Fourthly, the distribution of the $\ln RR$ effect sizes is non-normal (notably more so than the distribution of Hedges' g), perhaps because one can get extreme values when taking a ratio (e.g. if the denominator is close to zero, the ratio can be very large). Accordingly, Lajeunesse (2015) showed that $\ln RR$ is problematic when quantifying the outcome of primary studies with small sample sizes, and can yield unsuitable variance in some cases. So, Hedges' g has the possible practical advantage of increasing model fit in our meta-analysis.

2 — The authors may consider also doing another set of meta-analyses using lnVR (log variability ratio - proposed in Nakagawa et al. 2015).

Nakagawa, S., R. Poulin, K. Mengersen, K. Reinhold, L. Engqvist, M. Lagisz, and A. M. Senior. 2015. Meta-analysis of variation: ecological and evolutionary applications and beyond. *Methods in Ecology and Evolution* 6:143-152.

As one can see in Figure 2, the mean results (Hedges' g) are a mirror image of the variance results (this makes sense CV controls for means). I would like to see what the absolute change in variances. Probably the authors can put the analysis using lnVR in the supplement. The results of this analysis can be discussed. Also, the mean-variance relationship between mean and variance (sd) should be verified (e.g. plot log mean and log sd or log variance).

Many thanks – we now also present a meta-analysis of variation using lnVR in addition to one using lnCVR. We have the following comments on this new analysis:

Firstly, as the reviewer suggested, we now justify our use of lnCVR (as opposed to lnVR) by assessing the mean-variance relationship in our data. The major limitation of lnVR is that it does not account for the mean-variance relationship; indeed this is why lnCVR has been recommended over lnVR (Nakagawa et al. 2015). We indeed found a strong positive correlation between mean and variance, suggesting that we should favour lnCVR over lnVR.

Although we think that $\ln VR$ is not preferable due to this positive relationship, we ran our meta-analyses using $\ln VR$ in the interests of completeness. We present this new result in the supplementary material and briefly discuss it in the manuscript. To summarise, we found very similar patterns, although the statistical significance of some of the results has changed; see the following figure:

3 — I^2 needs to be explained. I^2 is proposed originally here:

Higgins, J., and S. Thompson. 2002. Quantifying heterogeneity in a meta-analysis. *Statistics in Medicine* 21:1539 - 1558.

But later expanded in here for mixed models (hierarchical models)

Nakagawa, S., and E. S. A. Santos. 2012. Methodological issues and advances in biological meta-analysis. *Evolutionary Ecology* 26:1253-1274.

This is what the authors use. Also, it will be good to put the degree in I^2 in the context. To do this see this paper:

Senior, A. M., C. E. Grueber, T. Kamiya, M. Lagisz, K. O'Dwyer, E. S. A. Santos, and S. Nakagawa. 2016. Heterogeneity in ecological and evolutionary meta-analyses: its magnitude and implications. *Ecology* 97:3293-3299.

Many thanks for this feedback – we have incorporated all these suggestions in the Methods section.

4 — Publication tests have been conducted on, I think, “meta-analytic” residuals results as suggested in Nakagawa and Santos 2012. One could use such residuals to conduct the trim and fill method and see how much the mean could move (see Nakagawa and Santos 2012). One should remember the funnel asymmetry could be caused by the presece of heterogeniety. See:

Egger, M., G. Smith, M. Schneider, and C. Minder. 1997. Bias in metaanalysis detected by a simple, graphical test. *Br Med J* 315:629 - 634.

Thanks – we now remind the reader that funnel plot asymmetry is not decisive evidence for publication bias, because it can also result from unexplained heterogeneity in the original effect sizes. We have elected not to implement trim-and-fill since the method seems to have fallen into disfavour, and because we are unsure how to correctly implement it using the REML- and Bayesian methods we used. NB that there is no method for trim-and-fill in mixed meta-analysis in the popular R package `metafor` (which we used), suggesting the method is not simple to implement.

5 — Figure 3 - are these grey envelopes 95% CI?

Yes – we have now included this information in the figure caption.

6 — the title - do the authors include “a systematic review” - so “a systematic reveiw and meta-analysis” - these two things are diferent - see:

Nakagawa, S., and R. Poulin. 2012. Meta-analytic insights into evolutionary ecology: an introduction and synthesis. *Evolutionary Ecology* 26:1085-1099.

We agree, and we have changed the title to “Sexual selection improves population fitness: a systematic review and meta-analysis”.

Hope my comments are useful.

Very useful, thank you!

Comments from Reviewer 3

This manuscript addresses the question of whether, on average, sexual selection has a net beneficial or detrimental effect on population fitness, using a meta-analysis of experimental evolution studies comparing the fitness of populations under different intensities of sexual selection. The analyses consider the effects of sexual selection on both the mean and variance of population fitness, and test whether these effects differ for stressful vs. benign environments, or depend on the measure of population fitness.

These are issues of longstanding interest, and as there is now quite a wealth of experimental evolution studies addressing these questions a meta-analysis synthesising their findings is timely.

Many thanks for your time and attention, and for the valuable feedback.

The results of this meta-analysis broadly concur with current theoretical predictions, and are likely to interest a wide readership. Across all studies, environment types and fitness measures there was a small positive effect of sexual selection on mean population fitness, although effect sizes varied with the fitness measure used. The positive effect of sexual selection on fitness was strongest for females in stressful environments, who also showed reduced variance in fitness under sexual selection.

The authors discuss these findings thoughtfully, although they focus more on the (weaker) effects found for ‘direct’ measures of fitness while somewhat neglecting the effects seen for ‘indirect’ fitness measures (e.g. of attractiveness, mating latency, lifespan, ejaculate traits). These potentially deserve more consideration, especially as many appear to be male-limited traits or measured mainly in males, that would appear to have a direct bearing on male mating success (if an indirect link to overall fitness), yet despite this the overall effect of sexual selection on male fitness is not significant.

Please see our earlier response to Prof. Radwan: we do not consider male mating success (and related traits) to be correlates of population fitness, which is why the results differ between our strict analysis and the one that includes all the traits. Additionally, we chose to spend more time writing about the “direct” effects because our main focus is on population fitness. We agree that our secondary results are interesting as well, but we wanted to keep the manuscript focused and concise. Also, there is no theoretical controversy over how sexual selection should affect traits like male mating success; by contrast, the relationship between sexual selection and population fitness is contentious.

I have only a few more specific comments:

Does your approach consider variation in the intensity of sexual selection exclusively on males? This is worth noting (e.g. statements such as that on line 98-99 might be amended to “Sexual selection on males significantly improved female fitness.”)

Yes, we now also stress that we consider sexual selection on males, as suggested (changes are made throughout the manuscript).

Experiments manipulating the sex ratio to alter sexual selection might simultaneously decrease sexual selection on males while increasing sexual selection on females. Would this be classified as reduced sexual selection in your analyses? I don’t think you included this aspect of study design as a moderator variable in any of your analyses – given that close to half of the effects you include come from “alternative manipulations” (line 70) perhaps you could test whether this affects effect size?

Some of the studies used the enforced monogamy design (as in Holland and Rice 2002), where one compares monogamous family groups with polyandrous ones (i.e. multiple males and one female), and so these manipulate sexual selection on males only, and eliminate sexual selection on females. Others instead manipulated the adult sex ratio (e.g. 3 males 1 female, 2 of each, or 3 females 1 male); these studies manipulate sexual selection on both sexes simultaneously. For the latter type, we considered the strength of sexual selection on males to be variable of interest, though it is confounded with the strength of sexual selection on females. To test if there was any difference in the effect sizes induced by these two types of manipulation, we now include ‘manipulation type’ (named “Enforced.Monogamy” in the Supplementary Material and R code) as a moderator variable: there was no effect in our preliminary model (see above comment to Reviewer 1 and the revised Table S10).

It seems possible that sexual selection on females, and not only on males, could affect population fitness – potentially even more directly than sexual selection on males. And might differing extents of sexual selection in each sex interact with the differences you saw in fitness measured in males vs. females?

It is possible that sexual selection on females affects population fitness, although to our knowledge no-one has considered or modelled this before. Also, we are not aware of any experiments that have manipulated the opportunity for sexual selection on females independently of any other factors (e.g. sexual selection on

males; see above comment). So, we have opted not to discuss or analyse this further, as we think the data do not yet allow for a robust test.

L111-118 It seems a little odd that there is no residual heterogeneity in your I^2 estimates. From the supplemental information it is not entirely clear to me how you have adapted the function that is under development in “metaAidR”, but can you double check that you have appropriately incorporated the residual variance into these estimates?

Many thanks for raising this. The original I^2 mistakenly did not include residual heterogeneity in the denominator when calculating proportion of heterogeneity for each random effect. The supplementary material has now been changed, with correctly derived I^2 values included in the revised manuscript.

L307-311 Please clarify here that $\ln CVR$ was calculated as the ratio $\ln(CV fitness_{SS}/CV fitness_{noSS})$

We now clarify this as suggested.

Fig. 3 Please explain the dashed red and black lines in the figure caption.

We now clarify the confidence intervals on the funnel plot as suggested.

Tables 1, 2 Please check – the parameter estimates for ‘Female sex’ and test for ‘Female > Male’ are bolded inconsistently between the REML and Bayesian models where these estimates are identical.

The revised manuscript no longer has this disparity between REML and Bayesian estimates. However we noticed that Bayesian estimates had slightly wider confidence intervals than REML models leading to differences in statistical significance (e.g. Table S6).

References

- Bonett, Douglas G. 2009. “Meta-Analytic Interval Estimation for Standardized and Unstandardized Mean Differences.” *Psychological Methods* 14 (3). American Psychological Association: 225.
- Lajeunesse, Marc J. 2015. “Bias and Correction for the Log Response Ratio in Ecological Meta-Analysis.” *Ecology* 96 (8). Wiley Online Library: 2056–63.
- Nakagawa, Shinichi, Robert Poulin, Kerrie Mengersen, Klaus Reinhold, Leif Engqvist, Malgorzata Lagisz, and Alistair M Senior. 2015. “Meta-Analysis of Variation: Ecological and Evolutionary Applications and Beyond.” *Methods in Ecology and Evolution* 6 (2). Wiley Online Library: 143–52.

Reviewers' Comments:

Reviewer #1:

Remarks to the Author:

The authors made a good effort to address my concerns. In particular, they now report a separate analysis addressing the effect of sexual selection manipulation on traits with direct effect on population fitness, which is the most relevant analysis for the question being asked. This effect has proved ambiguous (significant with REML, but with 95% CIs overlapping zero with Bayesian analysis), which, coupled with some evidence for publication bias (lines 147-151), makes the main conclusion of positive effect of selection on population fitness rather tentative. This interpretational uncertainty, however, is not properly reflected in the revised manuscript, eg. in the abstract (l. 12-13) and discussion (172-175; 249-251). Furthermore, recommendations for conservation biology seem premature in this context.

Sorry for not being clear with my point on random slopes, I meant the first option the authors identified, but (as I already indicated in my original review) I appreciate that dataset at hand may make this difficult. I'm OK with the way this comment was taken into account.

I appreciate the added text explaining that stronger response in females is counterintuitive (l. 195-222). However, it is still not clear what the authors mean to say about the nature of selection on females. I can see two options (which the authors also identify in their response to my question) (i) stronger correlated response in females to selection on males, compared to response in males themselves and (ii) direct effect of sexual selection manipulation on female fitness, e.g. due to increased male harassment. These two mechanisms should be clearly separated and discussed, they can have different consequences for interpretations of the results, but also on recommendations about allowing for "good genes" sexual selection in endangered species (eg. line 243).

Other comments:

l. 21: I wonder if defining population fitness as an average (presumably in both sexes) makes sense? As the authors state later in abstract, what really matters is fitness of demographically-limiting sex (and the traits which affect the demography).

162: low effect on sexual selection manipulation on traits like viability or female fecundity, compared to "indirect effect traits" which including male mating success, align with findings of meta-analysis of Prokop et al. 2012 – it would be appropriate to mention this consistency here

l. 165: but see recent metaanalysis by Kelly et al. (2018) showing no sex effect on immunity
I hope supplementary figures will be linked to the main text, as it is somewhat difficult to find them by the number in the supplement deposited in a repository

Hope this is useful, best regards

Jacek Radwan

Reviewer #2:

Remarks to the Author:

I read the point-to-point replies and the revised manuscript. I think that the authors did a very thorough job of addressing my comments as well as the other 2 reviewers' comments. This will be a nice contribution.

Reviewer #3:

Remarks to the Author:

The authors have thoughtfully responded to the comments of all reviewers, and I think they have

justified the details of their approach well. The changes they have made to the manuscript in this revision have clarified many of these details and help to better support the interpretation of results put forward in their discussion. I particularly appreciate the numerous additional analyses that are now presented as sensitivity tests of the main conclusions. I have no further comments for the authors.

Response to Prof. Jacek Radwan

1. *The authors made a good effort to address my concerns. In particular, they now report a separate analysis addressing the effect of sexual selection manipulation on traits with direct effect on population fitness, which is the most relevant analysis for the question being asked. This effect has proved ambiguous*

(significant with REML, but with 95% CIs overlapping zero with Bayesian analysis), which, coupled with some evidence for publication bias (lines 147-151), makes the main conclusion of positive effect of selection on population fitness rather tentative. This interpretational uncertainty, however, is not properly reflected in the revised manuscript, eg. in the abstract (l. 12-13) and discussion (172-175; 249-251). Furthermore, recommendations for conservation biology seem premature in this context.

We have modified the title and abstract of the paper, and made it clear which conclusions are supported unambiguously and which are more tentative. For example. . .

- The title is now *Meta-analytic evidence that sexual selection improves population fitness*.
- In the abstract we include: **We find evidence** that sexual selection on males tends to elevate the mean and reduce the variance for many fitness traits.
- In the abstract we include: However, sexual selection had weaker positive effects on direct measures of population fitness, such as extinction rate and proportion of viable offspring, than on individual fitness measures with indirect or less clear relationships to population fitness.
- We have modified the 1st paragraph of the discussion and make it clear which conclusions are more or less strongly supported by the data (see also following comments for additional changes to improve clarity of interpretation).

2. *Sorry for not being clear with my point on random slopes, I meant the first option the authors identified, but (as I already indicated in my original review) I appreciate that dataset at hand may make this difficult. I'm OK with the way this comment was taken into account.*

We are glad that the reviewer found the modifications we made satisfactory.

3. *I appreciate the added text explaining that stronger response in females is counterintuitive (l. 195-222). However, it is still not clear what the authors mean to say about the nature of selection on females. I can see two options (which the authors also identify in their response to my question) (i) stronger correlated response in females to selection on males, compared to response in males themselves and (ii) direct effect of sexual selection manipulation on female fitness, e.g. due to increased male harassment. These two mechanisms should be clearly separated and discussed, they can have different consequences*

for interpretations of the results, but also on recommendations about allowing for "good genes" sexual selection in endangered species (eg. line 243).

We agree that this section of the discussion can be clearer and have modified the two paragraphs in question.

The section in questions rests upon the fundamental premise that the response to selection depends on A) genetic variance, and B) the strength of selection. This fact is captured in the Breeder's equation, $R = h_2s$, where R is the response to selection, h_2 is the narrow-sense heritability of the trait, and s is the selection differential. Because females responded more strongly than males did (bigger R for females), our argument is that h_2 and/or s must be bigger for females. We then list a number of mechanisms that might make h_2 or s bigger for females.

Here are the revised paragraphs:

The results of the meta-analysis support predictions that sexual selection on males can improve population fitness and accelerate adaptation¹⁻⁷. One possible mechanism is that male mating success is positively genetically correlated with traits that contribute to population fitness, allowing females to benefit from a genome that has been purged of deleterious alleles through competition between males^{1,6,8}. A second (non-exclusive) mechanism is that experimental manipulation of sexual selection on males might

directly alter the selective pressures acting on females, causing female traits to evolve. For example, removing sexual selection via enforced monogamy probably alters selection on females, because it alters the frequency of interactions with males (as well as the evolved genotype of those males). What is less clear is why the manipulation of sexual selection had a larger effect on female trait means and variances as opposed to males – this result is arguably the opposite of what one would expect, since it is males that experience stronger sexual selection. Below we discuss possible explanations for this result, in light of the core principle that the extent of adaptation depends on additive genetic (co)variance and the strength of selection^{9,10}.

Firstly, it is possible that female traits show more additive genetic variation than male traits, causing female traits to respond more strongly to a change in selection. This hypothesis is plausible because males frequently do experience stronger selection than females¹¹, and sustained strong selection reduces heritability. A systematic review found no overall difference in mean heritability between male and female traits¹², but did record numerous instances in which trait heritability was higher for females than males¹². The sex chromosomes provide another reason for sex-specific heritability. Males are heterogametic in most of the species in our sample (*i.e.* the species with XY or XO sex determination), which can reduce father-to-son heritability relative to mother-to-daughter, since sons do not inherit the larger sex chromosome from their fathers^{13–16}, potentially slowing the adaptation of male traits^{16,17}.

Secondly, selection on males might be weaker than selection on females, resulting in slower adaptation following the experimental manipulation of sexual selection. This explanation may initially seem implausible, because net selection on males is often stronger than on females¹¹, due in part to the elevated importance of sexual selection in males as opposed to females^{18–21}. However, an oft-overlooked aspect is that selection might frequently be softer on males and harder on females²², because the local competitive environment is usually more important for males than it is for females. For instance, a mediocre male genotype can have high fitness provided it outcompetes its local rivals, while low-fitness female genotypes might produce few offspring even when competing with other low-fitness females. Therefore, improvements in genetic quality might have stronger diminishing returns in males, possibly contributing to our finding that the genetic consequences of sexual selection lead to greater fitness benefits for females. Though this argument is speculative, we note that many experimental evolution designs exaggerate the sex difference in the softness of selection, relative to expectations for large, natural populations^{23–25}. For example, many studies^{26–30} have evolved insects in small sub-populations, each containing one female and multiple males, whose progeny are then mixed and randomly sampled to create the next generation; this design ensures that successful males simply needed to outcompete their rival(s) in the same sub-population (soft selection), while each female's reproductive output is measured against the entire female population (hard selection).

Other Comments

4. I. 21: I wonder if defining population fitness as an average (presumably in both sexes) makes sense? As the authors state later in abstract, what really matters is fitness of demographically-limiting sex (and the traits which affect the demography). We have not made a change here, for two reasons.

Firstly, it usually makes no difference if one thinks of population fitness as an average across all the females, or an average across both sexes. This is because the average fitness of males is always equal to the average fitness of females, assuming the population has obligate sexual reproduction and a 50:50 primary sex ratio (as most species in our meta-analysis do), because every offspring has one genetic mother and one genetic father. For example, if the average fitness across all the females is 1.1, the average fitness of all the males must be 1.1 as well, so one gets the same answer when averaging over females only, or the whole population.

Secondly, 'population fitness' has a prominent technical definition that has been used by eminent figures in population geneticists (e.g. Wright, Fisher and Kimura), *i.e.* the average across the whole population. However, many of these classic models implicitly ignored males (e.g. by assuming isogamous sexual reproduction) for reasons of mathematical tractability. Later ecological and evolutionary models have incorporated the sexes explicitly, and unsurprisingly found that adaptation in males and females has

different effects on the growth rate of the population (confusingly, the growth rate of the population is also called 'population fitness' by many ecologists). We believe that a clear, inclusive, and universally satisfying definition of population fitness is currently missing, but it is beyond the scope of this paper – instead, we use the standard definition.

5. l. 162: low effect on sexual selection manipulation on traits like viability or female fecundity, compared to “indirect effect traits” which including male mating success, align with findings of meta-analysis of Prokop et al. 2012 – it would be appropriate to mention this consistency here.

Thank you – we have now added this reference to the sentence in question.

6. l. 165: but see recent metaanalysis by Kelly et al. (2018) showing no sex effect on immunity

This study is interesting, but ultimately we don't think it is sufficiently relevant to cite here. Our results found evidence that the experimental manipulation of sexual selection causes immunocompetence to evolve, while Kelly et al. (2018) examined whether males and females are differently defended against pathogens.

7. I hope supplementary figures will be linked to the main text, as it is somewhat difficult to find them by the number in the supplement deposited in a repository.

In the revision, we have provided a more traditional PDF version of the supplementary material, which is a simple list of all the supplementary figures and tables (this was also added to conform to Nature Communications style). For the submitted version of the manuscript we have hyperlinked the Supplementary Figure/Table to the relevant section of the Github page. One can also find a particular figure in the HTML version using Ctrl + F to search for the figure by name.